# FUSION-𝒳: ADVANCING LLM ABILITY WITH ADAPTIVE HETEROGENEOUS MODEL INTEGRATION

## ABSTRACT

Training LLMs presents significant challenges, including data access, privacy concerns, the complexity of training schedules, and limited resources. Therefore, a more accessible approach involves integrating existing LLMs, each tailored for different tasks or trained on distinct datasets, into an advanced and robust model with enhanced capabilities. Popular methods like ensemble and weight merging require substantial memory and struggle to adapt to changing data environments. Recent efforts have aimed to transfer only the collective knowledge of multiple LLMs to the target LLM. However, the resulting fused model often suffers from interference and performance degradation due to a lack of flexibility in the fusion process, including candidate selection and training pipeline. To address these issues, we propose a dynamic fusion framework to adaptively select LLMs for integration. Specifically, to diminish knowledge interference during LLM fusion, we introduce an adaptive selection network. It is a learnable mechanism that explicitly evaluates and selects the best-performing source LLMs based on their rewards, allowing us to fuse knowledge from a flexible number model candidates. To improve the knowledge fusion process, we propose a dynamic weighted fusion strategy that considers the intrinsic characteristics of candidate LLMs during fusion. Additionally, we incorporate a feedback-driven loss function to prevent the selector from converging to a state where it consistently assigns the same candidates. Our experiments demonstrate that our method consistently enhances model performance across multiple benchmarks, yielding an improvement of up to 2.2%. Additionally, our approach achieves a notable reduction in knowledge interference, showing up to 50% decrease compared to existing work.

## 1 INTRODUCTION

The emergence of large language models (LLMs) has led to the development of many specialized versions Zhang et al. (2023c); Christophe et al. (2024); Geng et al. (2023); Chenghao Fan & Tian (2023); Li et al. (2023), each tailored to specific domains and enhancing various aspects of daily life. However, the complexities involved in data collecting and pre-processing, training schedule design, and substantial training energy are challenging, particularly for academic researchers with limited experience and resources. Moreover, significant difficulties arise in fields like healthcare and business, where privacy concerns restrict data sharing between institutions. This limitation hampers the ability to gather sufficient data to train robust models. Consequently, institutions and companies often rely solely on their own datasets, which constrains their ability to develop superior models. Given these challenges, a more straightforward approach has gained attention Jiang et al. (2023); Jin et al. (2022); Zhang et al. (2023b); Wan et al. (2024a): instead of developing a new LLM from scratch, integrating existing LLMs into a unified model is more appealing. This integrated model serves as a robust base that can be further refined for various tasks, thereby avoiding the need for extensive individual training or data transfer.

Existing solutions, such as ensemble methods Jiang et al. (2023); Lu et al. (2023), enhance prediction performance by aggregating outputs from multiple models but require substantial memory and increase inference time due to the need to maintain and operate several models simultaneously. Another method involves merging several neural networks into a single network within the parameter space Jin et al. (2022). This generally presumes uniform network architecture and relies on manually configured weight merging or adding additional layers. Additionally, Mixture of Expert (MoE)

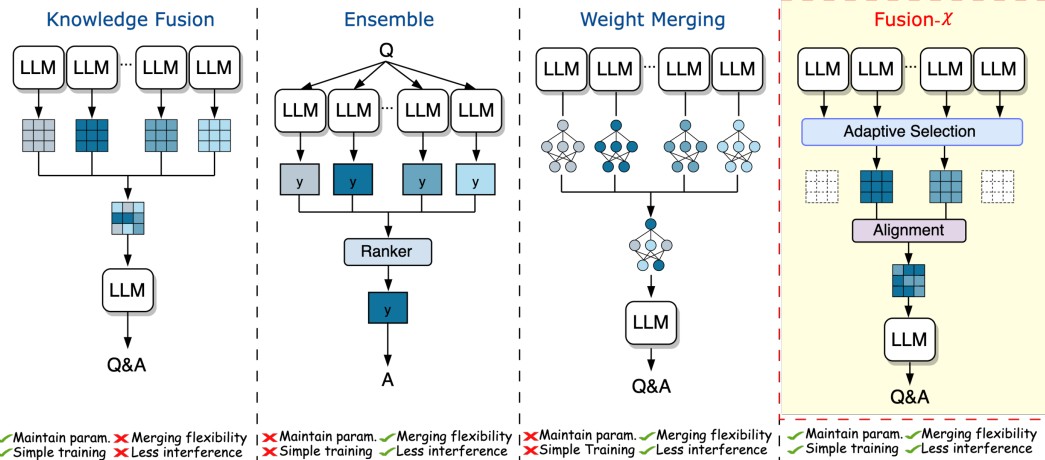

Figure 1: **Different model integration approaches** including ensemble, weight merging, knowledge fusion, and our method. We evaluate these methods based on four aspects: parameter size maintenance, simplified training, merging flexibility, and minimized interference.

structures Shazeer et al. (2017); Lepikhin et al. (2020); Du et al. (2022); Fedus et al. (2022), such as Mistral-7bx8 Jiang et al. (2024), address some inference and weight-sharing issues, but still suffer from long inference times and the requirement for homogeneous architectures and larger model sizes. FuseLLM Wan et al. (2024a) and FuseChat Wan et al. (2024b) have attempted to integrate the knowledge of multiple source LLMs using generated probability distribution matrices. However, these approaches suffer from interference and performance degradation in various tasks compared to the original target model due to unoptimized model selection and uncontrolled fusion processes. Fig. 1 provides an overview of the various model integration approaches. Our approach aims to address four critical challenges: *Parameter size maintenance*, *simplified training*, *merging flexibility*, and *minimized interference*, where existing methods fail to fully meet.

To overcome the limitations of existing LLM integration approaches, we propose a novel dynamic integration framework that adaptively selects LLMs for integration. More specifically, given a diverse set of random source LLMs with heterogeneous structures, we introduce an adaptive selection network, a learnable mechanism that explicitly evaluates and selects the best-performing source LLMs based on their rewards, thereby alleviating the interference issues typically caused by model fusion. The rewards are calculated based on each model's performance across a predefined set of tasks. Our framework allows flexibility in the number of LLMs selected during this process. To improve the knowledge fusion process, we implement a dynamic weighted fusion strategy that accounts for the intrinsic characteristics of candidate LLMs during fusion. The weights assigned in this process are derived from the reward evaluations, enabling the integration to favor models more likely to improve the overall performance of the composite LLM. The selector often converges to a state in which it consistently assigns large weights to the same few candidates. To address this, we incorporate a feedback-driven loss function that facilitates the training of our adaptive selection network and guides the selection of candidates.

Our method enhances the efficiency and effectiveness of LLM integration while maintaining adaptability and robustness against model diversity and data variability. It achieves this without increasing the parameter size as well as computation. of the target model, ensuring computational efficiency compared to traditional methods. Our contributions are as follows:

- We find that merely increasing the number of fusion candidates and expanding the source model pool does not necessarily enhance the fusion process, a selective strategy is more effective in minimizing knowledge interference.
- We propose a novel dynamic integration framework that adaptively selects LLMs for integration, leveraging an adaptive selection network, a permutation assignment fusion strategy, and a feedback-driven loss function to alleviate interference issues and enhance overall model performance.
- Our model outperforms existing approaches across multiple benchmarks, achieving a notable reduction in knowledge interference, with up to 50% decrease compared to previous methods.

## 2 RELATED WORK AND MOTIVATION

### 2.1 MODEL INTEGRATION

Research on model integration has evolved into distinct categories, each addressing different aspects of combining models: 1) *Ensemble:* LLM-Blender Jiang et al. (2023) uses ensemble techniques to enhance performance by combining outputs from multiple models. This process includes inferring all candidate models and then ranking them, which can be resource-intensive and slow. 2) *Weight Merging:* Zipit Stoica et al. (2023) merges partial layers of two models without additional training, creating a multi-head model for various tasks. Jin et al. (2022) merges models in their parameter space, guided by weights that minimize prediction differences. Rame et al. (2022); Arpit et al. (2022); Wortsman et al. (2022) employ weighted averaging methods. Zhang et al. (2023b) compose models through linear arithmetic operations in the weight space. These techniques are typically limited to models with identical architectures. 3) *Composition:* CALM Bansal et al. (2023) uses cross-attention mechanisms between models to integrate their representations, introducing new functionalities while maintaining each model's parameters, which can be inefficient in terms of speed and size. 4) *Knowledge Fusion:* FuseLLM Wan et al. (2024a) and FuseChat Wan et al. (2024b) focuses on fusing the probability distributions from various LLM candidates, integrating them into a single base LLM, blending knowledge across models. However, these approaches suffer from knowledge interference and performance degradation on various tasks compared to the original target LLM due to unoptimized model selection and uncontrolled fusion processes.

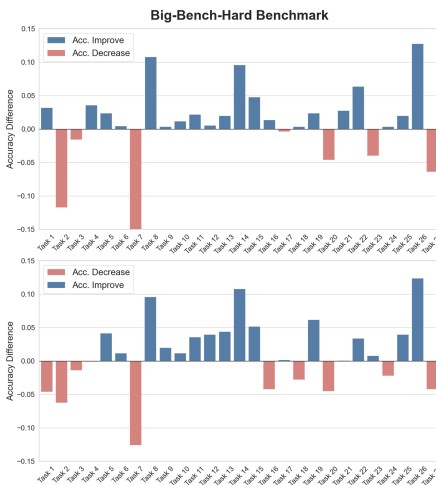

Figure 2: Evaluation on BBH benchmark when merging three models (upper) and four models (lower). The performance decreases on certain tasks for the integrated model, and the interference increases when integrating more models.

Knowledge distillation Hinton et al. (2015) is also used to integrate information into a model. However, student models are typically smaller and have lower performance than their teacher models. In our scenario, there is no limitation on the size or performance of the source models.

### 2.2 KNOWLEDGE INTERFERENCE

Knowledge Interference refers to the adverse impact on a model's performance when incorporating knowledge from other models, leading to decreased task performance. This can occur due to: 1) *Conflicting Information*: Models trained on different datasets or tasks contribute conflicting knowledge, resulting in confusion and degraded performance on specific tasks. 2) *Dilution of Valuable Knowledge*: Introducing less relevant or lower-quality information can dilute the original model's knowledge. 3) *Overfitting to Irrelevant Patterns*: The fused model may overfit to noise or less useful patterns from new models, losing focus on critical aspects of the tasks it was initially trained on. Fig. 2 shows the results of FuseLLM Wan et al. (2024a) on 27 tasks of the Big-Bench Hard Suzgun et al. (2022) benchmark, exhibiting performance degradation on multiple tasks (in red). We also observe that fusing more models (upper part: fusing 3 models, lower part: fusing 4 models) can lead to even more degradation. Detailed results are shown in Tab. 3.

## 3 PRELIMINARIES

As a general integration approach parallel to ensembling and weight merging, knowledge fusion combines the probabilistic distribution matrices $P_i^{\theta_i}$ from multiple LLMs. These matrices reflect each model's inherent knowledge for text understanding. Let $t$ be a text sequence of length $N$, and $t_{<i}$ denote the sequence preceding the $i$-th token. The probabilistic distribution matrix $P_t^{\theta_i}$ for the

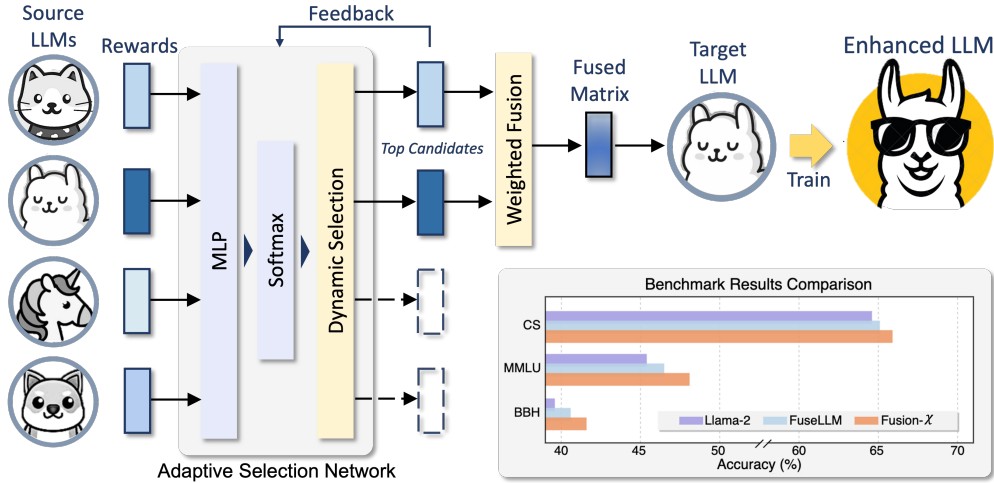

Figure 3: **Overall framework:** Multiple source LLMs are evaluated and selected based on performance by an adaptive selection network. Top candidates then proceed through a dynamic weighted fusion process guided by a feedback loss to enhance the ability of the target LLM. The lower right shows evaluation results on CommonSense, MMLU, and Big-Bench-Hard benchmark.

$i$-th LLM is given by:

$$P_t^{\theta_i} = [p_\theta(t_1|t_{<1}), p_\theta(t_2|t_{<2}), \ldots, p_\theta(t_N|t_{<N})], \tag{1}$$

where, $P_t^{\theta_i}$ is the probabilistic distribution matrix for the $i$-th LLM for text sequence $t$. $p_{\theta_i}(t_k|t_{<k})$ is the predicted probability distribution for the $k$-th token given the preceding tokens $t_{<k}$, according to the $i$-th LLM parameterized by $\theta_i$. Each element $p_{\theta_i}(t_k|t_{<k})$ is a vector of probabilities corresponding to each token in the vocabulary, summing to 1.

Their fusion process is achieved by minimizing the divergence between the target LLM's (pre-defined among the source LLMs) probabilistic distributions and those of the source LLMs:

$$\boldsymbol{P}_f = \mathcal{F}(P_1^{\theta_1}, P_2^{\theta_2}, \ldots, P_N^{\theta_N}), \tag{2}$$

where $\mathcal{F}$ denotes the function that combines multiple matrices. $\{P_i^{\theta_i}\}_{i=1}^N$ represents the representation matrix of each LLM. We simplify the notation as $P_i$ in the later equations.

The overall objective for continual training consists of a weighted combination of the causal language modeling objective $\mathcal{L}_{lm}$ and the fusion objective $\mathcal{L}_{fuse}$ as follows:

$$\mathcal{L} = \lambda \mathcal{L}_{lm} + (1 - \lambda)\mathcal{L}_{fuse}, \tag{3}$$

where $\mathcal{L}_{lm}$ is the causal language modeling objective, and $\mathcal{L}_{fuse}$ is the cross-entropy loss between the target LLM's predictions (output) and the fused representation matrix $P_f$.

## 4 METHODOLOGY

Our method advances existing knowledge fusion methods by introducing a dynamic framework that consists of an Adaptive Selection Network and a Dynamic Weighted Fusion mechanism, as illustrated in Fig. 3. Specifically, at each training step, the Adaptive Selector evaluates performance metrics to dynamically select a subset of candidate models based on their probabilistic distribution matrices rather than all candidates. More importantly, both the selection of candidates and the number of candidates selected are adaptive, preventing knowledge interference and enhancing overall model performance. The selected candidates are then fused using a weighted sum based on normalized selection probabilities. This process is guided by a specially designed loss function that refines model selection through feedback. Our framework provides flexibility for future scalability and allows the integration process to accommodate varying computational constraints and application needs.

### 4.1 ADAPTIVE SELECTION NETWORK

We propose an Adaptive Selection Network (ASN) to evaluate the source models based on a continuous learning process. It integrates feedback from ongoing interaction, which will be introduced in

Table 1: **Experiments on Different Design Choices** of our framework under Fusion-$\mathcal{X}$-T scale with 4 source models. We show ablation results on Commonsense and BBH, along with perplexity results.

(a) Selection count. Adaptively selecting the candidates outperforms selecting the top 2 or all of them.

| Method | PPL↓ | CS↑ | BBH↑ |
|---|---|---|---|
| Top-2 | 11.67 | 40.55 | 6.75 |
| Adaptive | **11.04** | **41.32** | **7.31** |
| All | 11.91 | 40.52 | 6.64 |

(b) Layer Choice. Adding 3 linear layers for our selector leads to the best performance.

| Method | PPL↓ | CS↑ | BBH↑ |
|---|---|---|---|
| Conv. | 12.21 | 39.73 | 6.11 |
| $1 \times$ Linear | 11.42 | 40.78 | 6.86 |
| $3 \times$ Linear | **11.04** | **41.32** | **7.31** |

(c) Adding Loss. Adding additional feedback loss leads to better results.

| Method | PPL↓ | CS↑ | BBH↑ |
|---|---|---|---|
| w/o Loss | 11.48 | 40.91 | 6.92 |
| Feed. loss | **11.04** | **41.32** | **7.31** |

(d) Fusion method. Averaging candidates weighted by their reward ensures best result.

| Method | PPL↓ | CS↑ | BBH↑ |
|---|---|---|---|
| Avg | 11.32 | 40.85 | 6.80 |
| Max | 11.77 | 40.11 | 6.68 |
| w/o Weighted | 11.96 | 39.82 | 6.38 |
| Weighted | **11.04** | **41.32** | **7.31** |

(e) Threshold setting. We show the average selected candidates of each threshold.

| Weight | Avg. | PPL↓ | CS↑ | BBH↑ |
|---|---|---|---|---|
| 0.2 | 1.27 | 13.78 | 39.24 | 5.07 |
| 0.15 | 2.82 | **11.04** | **41.32** | **7.31** |
| 0.12 | 3.64 | 11.67 | 40.65 | 6.75 |
| 0.1 | 4.00 | 11.91 | 40.52 | 6.64 |

(f) Selection metric. Using softmax outperforms gumbel softmax or adding additional random noise.

| Method | PPL↓ | CS↑ | BBH↑ |
|---|---|---|---|
| Softmax | **11.04** | **41.32** | **7.31** |
| Gumbel | 13.41 | 39.15 | 5.00 |
| Noise | 13.11 | 38.97 | 4.90 |

Sec. 4.3. Various methods are explored during the design process, as shown in Tab. 1. The network takes the normalized matrices $P_i$ (regarded as rewards) as input, which are flattened and normalized using layer normalization to stabilize training. It then computes the logits for each candidate model. The network consists of three linear layers (see Tab. 1b) with specified dimensions and uses the GELU activation function for enhanced non-linearity, improving the network's ability to capture complex patterns in the input data. The logits for any $P_i$ from the $\{P_i^{\theta_i}\}_{i=1}^{N}$ can be defined as the following expression:

$$z_\phi(P_i) = (f^3 \circ \text{GELU} \circ f^2 \circ \text{GELU} \circ f^1)(P_i), \tag{4}$$

where $f^1$, $f^2$, and $f^3$ represent the linear layers. The adaptive selection mechanism utilizes the scores from the network, converting these into a probability via the softmax function:

$$p_i = \frac{e^{z_\phi}}{\sum_{i=1}^{N} e^{z_\phi}}, \tag{5}$$

where $p_i$ is the softmax probability associated with the $i$-th candidate, and $N$ is the total number of candidates. We also compared the effects of adding Gumbel softmax Jang et al. (2016) or noise Shazeer et al. (2017) before the softmax (see Tab. 1f). The better performance of softmax shows that the selection process benefits more from the smooth and differentiable mapping of logits, as well as improved convergence, rather than from adding randomness and increasing variance.

**Dynamic Candidate Selection**  To determine which candidate models to select for fusion, we apply a dynamic thresholding mechanism. A threshold $\tau$ is set, and candidates with selection probabilities exceeding this threshold are selected:

$$\mathcal{X}_{\text{selected}} = \left\{ P_j^{\theta_j} \mid p_j > \tau, \, j = 1, \ldots, K \right\}, \tag{6}$$

where the output set $\mathcal{X}_{\text{selected}} = \{P_j^{\theta_j}\}_{j=1}^{K}$ represents a subset of the original set of models $\{P_i^{\theta_i}\}_{i=1}^{N}$. We simplify the notation $P_j^{\theta_j}$ as $P_j$ in the later equations.

To ensure that at least one candidate is selected per sample $1 \leqslant K \leqslant N$, we check if no candidates meet the threshold and, if so, select the candidate with the highest probability:

$$\text{If } |\mathcal{X}_{\text{selected}}| = 0, \text{ then } \mathcal{X}_{\text{selected}} = \{\arg\max_j p_j\}. \tag{7}$$

In our implementation, we set $\tau = 0.15$ (see Tab. 1e). This dynamic selection allows the model to adaptively choose the most relevant candidates (see Tab. 1a) based on the input data and current learning context.

## 4.2 DYNAMIC WEIGHTED FUSION

We proceed with the fusion process after selecting the candidate models. First, we normalize the weights of the selected probabilities obtained from Eq. 5:

$$\hat{p} = \frac{p \odot m}{\sum_{i=1}^{N} p_j m_j + \epsilon},$$

(8)

where $m_j$ is a binary mask indicating the selected candidates ($m_j = 1$ if $j \in \mathcal{X}_{\text{selected}}$, else $m_j = 0$), and $\epsilon$ is a small constant to prevent division by zero. To perform the weighted sum, we reshape the normalized probabilities and masks to match the dimensions of the candidate outputs, enabling element-wise multiplication. The outputs of the $K$ selected candidates $P_j$ are accumulated based on their respective weights to produce a unified model output $P_f$. This is calculated as follows:

$$P_f = \max\left(\texttt{concat}\left(\{P_j \cdot \hat{p}_j \cdot m_j\}_{j=1}^{K}, \texttt{dim=-1}\right), \texttt{dim=-1}\right),$$

(9)

We assign the proportion of the candidates' probabilistic distributions based on the weights in Eq. 8. $\texttt{concat}$ denotes the concatenation operation of all $K$-selected candidates. $\max$ function is applied for fusing the maximum value from these aligned metrics $P_f$. We found this dynamic fusion process can constantly let the more influential candidates have a greater effect on the final model (see Tab.1d). Next, the fused representation $P_f$ goes through the cross-entropy loss $\mathcal{L}_{\text{fuse}}$ in Eq. 3.

Our method fundamentally transforms the approach to integration by utilizing a data-driven, adaptive mechanism to dynamically evaluate contributions of candidate LLMs and select accordingly.

## 4.3 LOSS AND TRAINING PIPELINE

The selection network often converges to a state where it consistently assigns large weights to the same few candidates. To facilitate the training of our network, we implement a feedback approach to guide the selection of candidates (see Tab.1c). Consequently, we adopt a soft constraint approach. The importance of a model relative to a batch of training examples is defined as the batch-wise sum of the values $\hat{p}_j$ for each LLM. We define a feedback loss $L_{\text{feed}}$, which is added to the overall loss function for the model as described in Eq. 3. This loss is calculated as the square of the coefficient of variation $\mathcal{CV}^2$ of the importance values. The importance values are derived from the weights of different candidates in the model, summed over the index set $K$. This formulation is given by:

$$\mathcal{L}_{\text{feed}} = \mathcal{CV}^2\left(\sum_{j \in K} \hat{p}_j\right) = \frac{\sigma^2\left(\sum_{j \in K} \hat{p}_j\right)}{\mu^2\left(\sum_{j \in K} \hat{p}_j\right) + \epsilon},$$

(10)

Here, $\sigma^2$ is the variance, $\mu$ is the mean, and $\epsilon$ is a small constant added to ensure numerical stability (preventing division by zero). This refined definition emphasizes the goal of making the distribution of source LLMs' importance more uniform across the model. Minimizing the variance of the importance values $\hat{p}_j$ reduces the spread or difference between these values, making the distribution of importance more uniform. Simultaneously maximizing the mean ensures that the feedback loss does not become excessively sensitive to small variances. Squaring the mean in the denominator helps to normalize the loss and maintain a consistent scale, emphasizing relative changes in the variance. The full training objective is a combination of the above objectives:

$$\mathcal{L}(\theta_t, \phi_{ASN}) = \underbrace{-\mathbb{E}_{t \sim \mathcal{D}}\left[\mathcal{D}(P_t, O_t)\right]}_{\mathcal{L}_{\text{lm}}} + \underbrace{\lambda_{\text{fuse}}\left(-\mathbb{E}_{t \sim \mathcal{D}}\left[\mathcal{D}(P_t, P_f)\right]\right)}_{\mathcal{L}_{\text{fuse}}} + \underbrace{\lambda_{\text{feed}}\mathcal{CV}^2\left(\sum_{j \in K} \hat{p}_j\right)}_{\mathcal{L}_{\text{feed}}},$$

(11)

where $\mathcal{L}_{\text{lm}}$ reduces the discrepancy between $P_t$ and the one-hot label matrix, $O_t \in \{0, 1\}^{L \times D}$. $\theta_t, \phi_{ASN}$ are parameters of target LLM and selection network. $\mathcal{L}_{\text{fuse}}$ enforces assignment between the target LLM's predictions $P_t$ and the fused representation matrix $P_f$. We set $\lambda_{\text{fuse}} = 0.1$, $\lambda_{\text{feed}} = 0.5$ in our experiments. Grid search results are shown in Appx. C. Our method is described in Alg. 1.

---

**Algorithm 1** Fusion-$\mathcal{X}$ for LLMs Integration

---

**Require:** Source LLMs probabilistic distribution matrices $\{P_i\}_{i=1}^N$, training corpus $C$.
**Ensure:** Fused representation matrix $P_f$, Target LLM $\mathcal{T}$
 1: Initialize the adaptive selection network: $z_\phi(P_i)$.
 2: **for** each text in $C$ **do**
        `// Step1: Select fusion candidates with ASN.`
 3:    **for** each input $P_i$ **do**   `# Tensor shape:(L, D, N)`
 4:        Obtain logits $z_\phi(P_i)$ using Eq. 4.   `# Tensor shape:(LD, N)`
 5:        Calculate softmax probability $p_i$ using Eq. 5.
 6:    **end for**
        `// Step2: Fuse selected candidates using permutation assignment.`
 7:    Obtain $\mathcal{X}_{\text{selected}}$ using Eq. 6.   `# Selecting based on adaptive threshold τ`
 8:    Compute $P_f$ using Eq. 9.   `# Tensor shape:(L, D, K)`
        `// Step3: Training schedule.`
 9:    Calculate feedback loss $\mathcal{L}_{\text{feed}}$ using Eq. 10.
10:    Compute final loss $\mathcal{L}$ using q. 11   `# Combination of ℒₗₘ, ℒfᵤₛₑ, and ℒfₑₑd`
11:    Update model parameters based on it.
12: **end for**
13: **return** Trained $\mathcal{T}$.

---

## 5 EXPERIMENTS

### 5.1 IMPLEMENTATION DETAILS

**Models and Datasets.** We conduct experiments on various scales of models, including Llama-160M Miao et al. (2023), GPT-Neo-125M Black et al. (2021), Pythia-160M Biderman et al. (2023), Tiny-starcoder Li et al. (2023), LiteLlama-460M-1T, OpenLLaMA-V2-3B Geng & Liu (2023), MiniMA-3B Zhang et al. (2023a), Amber Liu et al. (2023), Starcoder2-3B Li et al. (2023), Llama-2-7B Touvron et al. (2023), OpenLLaMA-7B Geng & Liu (2023), Starcoder2-7B Li et al. (2023). These models have different parameter sizes, architectures, tokenizers, and vocabulary. We process model integration by first selecting the source models, then setting one model as our target model, and performing model fusion by fusing the representation matrices. Results are shown in Sec. 5.2. We follow Wan et al. (2024a) to use MiniPile Kaddour (2023) for continual training.

**Training details.** Our model is optimized using the AdamW optimizer with beta1 = 0.9 and beta2 = 0.95, with gradient clipping set to 1.0 and weight decay to 0.1. A cosine learning rate schedule is employed, with a maximum learning rate of 3e-5 for models under 1B and 1e-5 for models larger than 1B and a warmup ratio of 0.008. We train with 4 A100 GPUs, each with 80GB of memory.

**Evaluation benchmarks.** We evaluate Fusion-$\mathcal{X}$ on three benchmarks that represent different core capabilities of LLMs: Common Sense (CS) Talmor et al. (2018), Big-Bench Hard (BBH) Suzgun et al. (2022), Multi-task Language Understanding (MMLU) Hendrycks et al. (2021), and MultiPL-E (ME) Cassano et al. (2023), representing the ability of commonsense, reasoning, and code generation, respectively. A detailed description of the four benchmarks are presented in the Appx. D.

### 5.2 MAIN RESULTS

Tab. 2 shows the zero-shot performance of Fusion-$\mathcal{X}$ and the baseline methods on the Common Sense (CS) benchmark when fusing 4 LLMs. We present our model in three scales: 1) Fusion-$\mathcal{X}$-T: Integrated with Llama-160M, GPT-Neo-125M, Pythia-160M, Tiny-starcoder. 2) Fusion-$\mathcal{X}$-S: Integrated with OpenLLaMA-V2-3B, MiniMA-3B, Amber, Starcoder2-3B. 3) Fusion-$\mathcal{X}$-B: Integrated with Llama-2-7B, OpenLLaMA-7B, MPT-7B, Starcoder2-7B.

The results demonstrate that our model consistently surpasses the target models across all six tasks achieving an average performance improvement of 0.78% over Llama-160M, 1.07% over OpenLLaMA-3B, and 1.16% over Llama-2-7B , with a standard deviation of $-0.02 \sim +0.02$. Our model also outperforms the state-of-the-art fusion method, FuseLLM, at all scales. More

Table 2: **Overall results** of Fusion-$\mathcal{X}$ and baselines in commonsense evaluations on CommonSense (CS), where percentages indicate the rate of improvement/decrease compared to our target model, denoted with "*". Larger values indicate better results.

| Model / Task | ARC-easy | ARC-challenge | BoolQ | HellaSwag | OpenBookQA | Winogrande | Avg. 6 Tasks |
|---|---|---|---|---|---|---|---|
| Llama-160M* | 43.35 | 23.04 | 61.44 | 35.23 | 30.00 | 50.20 | 40.54 |
| GPT-Neo-125M | 43.60 | 22.95 | 61.68 | 30.44 | 26.20 | 50.67 | 39.26 |
| Pythia-160M | 43.90 | 23.55 | 54.59 | 30.24 | 27.00 | 51.38 | 38.44 |
| Tiny-starcoder | 30.72 | 20.31 | 61.68 | 29.24 | 25.20 | 51.78 | 36.49 |
| FuseLLM | 43.54 (+0.19) | 21.93 (-1.11) | 61.48 (+0.04) | 34.74 (-0.49) | 30.20 (+0.20) | 51.23 (+1.03) | 40.52 (-0.02) |
| Fusion-$\mathcal{X}$-T | 44.23 (+0.88) | 22.95 (-0.09) | 61.59 (+0.15) | 35.47 (+0.24) | 31.60 (+1.60) | 52.09 (+1.89) | 41.32 (+0.78) |
| OpenLLaMA-V2-3B* | 63.30 | 36.35 | 65.44 | 69.93 | 37.80 | 63.22 | 56.01 |
| MiniMA-3B | 25.88 | 28.41 | 62.17 | 25.19 | 28.20 | 49.33 | 36.53 |
| Amber | 65.87 | 36.60 | 68.72 | 72.41 | 41.40 | 64.33 | 58.22 |
| Starcoder2-3B | 55.47 | 30.80 | 64.40 | 46.43 | 30.00 | 54.70 | 46.97 |
| FuseLLM | 63.72 (+0.42) | 35.55 (-0.80) | 66.51 (+1.07) | 70.23 (+0.30) | 37.10 (-0.70) | 63.59 (+0.37) | 56.17 (+0.16) |
| Fusion-$\mathcal{X}$-S | 65.03 (+1.73) | 36.43 (+0.05) | 67.31 (+1.78) | 70.75 (+0.82) | 38.25 (+0.45) | 64.69 (+1.47) | 57.08 (+1.07) |
| Llama-2-7B* | 74.58 | 46.33 | 77.71 | 76.00 | 44.20 | 69.30 | 64.69 |
| OpenLLaMA-7B | 69.70 | 41.38 | 72.29 | 74.50 | 40.80 | 65.82 | 60.75 |
| MPT-7B | 70.12 | 42.15 | 74.74 | 76.25 | 42.40 | 68.15 | 62.30 |
| Starcoder2-7B | 60.61 | 34.90 | 69.08 | 51.00 | 32.00 | 55.17 | 50.46 |
| FuseLLM† | 75.04 | 47.44 | 78.13 | 76.78 | 45.40 | 69.03 | 65.30 |
| FuseLLM | 75.23 (+0.65) | 47.14 (+0.81) | 78.22 (+0.51) | 76.40 (+0.40) | 44.34 (+0.14) | 69.22 (-0.08) | 65.09 (+0.40) |
| Fusion-$\mathcal{X}$-B | 75.46 (+0.88) | 47.50 (+1.17) | 78.86 (+1.15) | 76.97 (+0.97) | 46.02 (+1.82) | 70.33 (+1.03) | 65.85 (+1.16) |

importantly, our method notably reduces knowledge interference during model integration, especially on tasks such as ARC-Challenge, HellaSwag, and OpenBookQA. Our approach effectively prevents model performance degradation caused by integrating models with less relevant or lower-quality information. Given that the source models have large differences in performance, our method ensures the preservation of the original model's knowledge. We will discuss this further in Appx. A.

Fig. 4 shows the results for the code generation evaluation, the zero-shot performance of Llama-2, FuseLLM, and our Fusion-$\mathcal{X}$ on the MultiPL-E (ME) benchmark. We observe that Fusion-$\mathcal{X}$ outperforms FuseLLM in all tasks. Our model surpasses Llama-2 by a large margin, with an average performance increase of remarkable performances in code generation tasks compared to Llama-2, the fusion result via Fusion-$\mathcal{X}$ achieves an average performance gain of 2.15%, which is higher than the 0.69% improvement observed in FuseLLM. Detailed results of multiple scales can be found in the Appx. G.

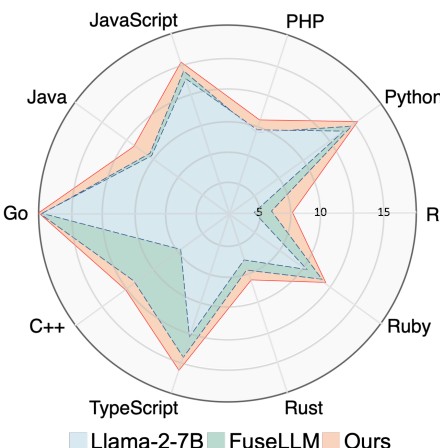

Figure 4: **Evaluation results** comparison on MultiPL-E benchmark.

The overall results of the Fusion-$\mathcal{X}$ model compared to baseline methods on the BBH benchmark are presented in Tab. 3. The four source LLMs show varying performance across the 27 BBH tasks. After integration, our Fusion-$\mathcal{X}$ model achieves an average improvement of 2.11% across all tasks, demonstrating the effectiveness of our approach. Compared to FuseLLM, our method doubles the performance increase (1.05 vs. 2.11) compared to Llama-2. We can also observe knowledge interference in some tasks. This interference arises due to two primary reasons. First, other source LLMs, apart from Llama-2, perform poorly on certain tasks, affecting the fusion results. Second, the relevance between the continual training dataset and downstream tasks contributes to performance degradation. Despite FuseLLM showing an average performance gain compared to Llama-2-7B, it performs worse than Llama-2 on 10 tasks, indicating significant knowledge interference. For instance, in the Snarks task, Llama-2 achieves 50.56%, while FuseLLM scores 46.21%. This drop is because all other LLMs perform worse on this task compared to Llama-2, highlighting how less relevant or lower-quality information can degrade the model's performance.

In contrast, Fusion-$\mathcal{X}$ only has 5 tasks that perform lower than Llama-2-7B, showing a 50% reduction in knowledge interference compared to FuseLLM. We are also able to reduce the performance drop of the tasks that are effected by knowledge interference. These results indicate that our method effectively limits knowledge interference, resulting in more consistent performance improvements.

Table 3: **Overall results** of Fusion-$\mathcal{X}$ and baselines in reasoning evaluations on Big-Bench Hard (BBH), where percentages indicate the rate of improvement/decrease compared to Llama-2-7B.

| Task | Llama-2* | OpenLlama | MPT | Starcoder | FuseLLM | Fusion-$\mathcal{X}$ |
|---|---|---|---|---|---|---|
| Boolean Expressions | 69.60 | 75.60 | 66.20 | 75.60 | 65.00 (-4.60) | 72.60 (+3.00) |
| Causal Judgement | 52.94 | 54.55 | 50.60 | 20.26 | 46.67 (-6.27) | 51.20 (-1.74) |
| Date Understanding | 62.80 | 43.20 | 43.60 | 51.20 | 61.40 (-1.40) | 57.60 (-5.20) |
| Disambiguation QA | 46.40 | 36.80 | 47.60 | 44.40 | 46.30 (-0.10) | 50.40 (+4.00) |
| Dyck Languages | 6.00 | 6.00 | 5.20 | 23.20 | 10.20 (+4.20) | 7.60 (+1.60) |
| Formal Fallacies | 49.60 | 50.80 | 52.40 | 41.60 | 50.80 (+1.20) | 50.20 (+0.60) |
| Geometric Shapes | 32.80 | 0.00 | 0.00 | 22.40 | 20.20 (-12.60) | 22.00 (-10.8) |
| Hyperbaton | 51.60 | 66.80 | 53.21 | 72.40 | 61.20 (+9.60) | 58.00 (+6.40) |
| Logical Deduction (3 objects) | 56.00 | 41.60 | 40.80 | 52.80 | 58.00 (+2.00) | 56.40 (+0.40) |
| Logical Deduction (5 objects) | 32.00 | 25.20 | 31.20 | 42.40 | 33.20 (+1.20) | 32.40 (+0.40) |
| Logical Deduction (7 objects) | 24.00 | 16.80 | 18.40 | 41.20 | 27.60 (+3.60) | 24.40 (+0.40) |
| Movie Recommendation | 70.40 | 40.00 | 52.00 | 45.20 | 74.40 (+4.00) | 72.80 (+2.40) |
| Multistep Arithmetic Two | 0.40 | 0.80 | 0.40 | 24.80 | 4.80 (+4.40) | 3.20 (+2.80) |
| Navigate | 53.20 | 53.20 | 48.80 | 75.20 | 64.00 (+10.8) | 63.60 (+10.4) |
| Object Counting | 49.20 | 49.20 | 40.40 | 52.40 | 54.40 (+5.20) | 54.80 (+5.60) |
| Penguins in a Table | 31.51 | 26.03 | 28.08 | 50.68 | 27.27 (-4.24) | 31.51 (+0.00) |
| Reasoning about Colored Objects | 48.00 | 28.00 | 31.60 | 57.20 | 48.20 (+0.20) | 52.00 (+4.00) |
| Ruin Names | 33.20 | 28.00 | 23.20 | 43.60 | 30.40 (-2.80) | 34.00 (+8.00) |
| Salient Translation Error Detection | 24.80 | 16.00 | 0.00 | 33.60 | 31.00 (+6.20) | 30.00 (+5.20) |
| Snarks | 50.56 | 44.38 | 45.51 | 40.01 | 46.21 (-4.35) | 54.44 (+3.88) |
| Sports Understanding | 88.40 | 66.00 | 82.40 | 53.60 | 88.50 (+0.10) | 90.40 (+2.00) |
| Temporal Sequences | 12.40 | 31.60 | 20.80 | 26.80 | 15.80 (+3.40) | 18.00 (+5.60) |
| Tracking Shuffled Obj. (3 objects) | 32.40 | 35.60 | 30.40 | 37.20 | 33.20 (+0.80) | 33.60 (+1.20) |
| Tracking Shuffled Obj. (5 objects) | 17.60 | 20.00 | 14.60 | 33.60 | 15.40 (-2.20) | 14.80 (-2.80) |
| Tracking Shuffled Obj. (7 objects) | 10.80 | 10.80 | 2.00 | 17.60 | 14.80 (+4.00) | 24.40 (+13.6) |
| Web of Lies | 51.60 | 53.20 | 63.60 | 57.60 | 61.80 (+10.2) | 60.00 (+8.40) |
| Word Sorting | 10.80 | 5.20 | 6.80 | 25.60 | 6.60 (-4.20) | 5.60 (-5.20) |
| **Avg. 27 Tasks** | 39.59 | 34.27 | 33.33 | 44.19 | 40.64 (+1.05) | 41.70 (+2.11) |

## 6 ABLATION & ANALYSIS

**Distribution of Activation Frequencies**

Fig. 3 provides an analysis of the LLM fusion candidate selection distribution when training Fusion-$\mathcal{X}$-T. The left panel shows the selection across 120K training steps, the consistent patterns observed in the selection distribution indicate that our adaptive selection network is effectively distributing the selection load among the candidates. The adaptive mechanism allows the network to dynamically adjust which LLMs to select based on the ongoing learning process.

The right panel illustrates the proportion of each selection throughout the training. The proportions are as follows 27.12%, 25.42%, 16.95%, 30.51%. These results indicate that our method finds LLM 4 (Tiny-starcoder) to be more valuable than the others, and LLM 3 (Pythia-160M) to be less valuable for the cur-

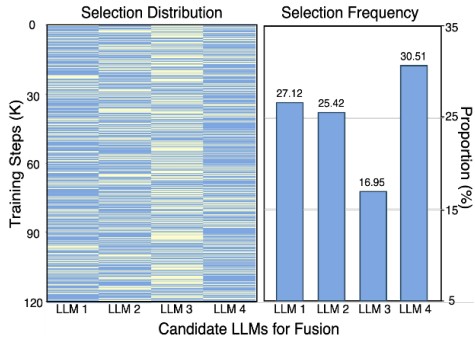

Figure 5: **Candidate selection distribution.** The Left shows the selection for each training step, and the right shows the proportion of each selection for the training.

rent integration process. Our statistical results demonstrate that we can accurately identify effective LLM candidates for the current task from the source model candidate pool at each training step.

**Scaling Results**

Model scaling is a critical evaluation for LLMs. In our paper, we conduct experiments in two directions: scaling the model size and scaling the number of source models. We evaluate model scaling on the Commonsense, BBH, and MMLU datasets, assessing both the average performance across all tasks and the knowledge interference on individual tasks. Results on the Commonsense dataset are shown in Tab. 2. For BBH and MMLU, we present the results in Fig. 6. In the figure, the lines represent the average performance (left y-axis) of FuseLLM and our model when fusing 3 LLMs,

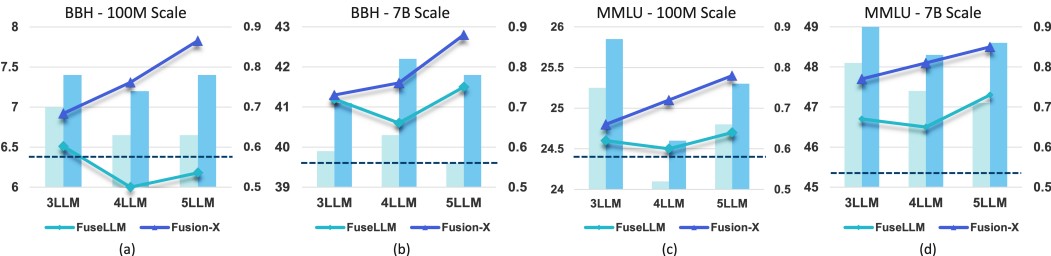

Figure 6: **Scaling number of fusion candidates.** The line charts display the accuracy for FuseLLM and our model on the BBH and MMLU benchmarks. The histograms show the percentage of tasks outperforming the baseline when integrating 3, 4, and 5 LLMs. Dotted lines represent the baseline.

4 LLMs, and 5 LLMs. The dotted line in each subfigure shows the baseline model performance of Llama-160M (160M scale) and LLaMA-2-7B (7B scale). In Fig. 6a, the performance of FuseLLM is even lower than the baseline on the BBH benchmark when fusing 4 LLMs and 5 LLMs. In contrast, our model consistently increases performance when integrating more LLMs. The performance degradation in FuseLLM is due to knowledge interference, which is reflected in the histogram (right y-axis). It shows the percentage of tasks BBH (total 27 tasks) and MMLU (total 57 tasks) that perform higher than the baseline. FuseLLM shows much less improvement compared to our model. Especially for the 4 LLMs and 5 LLMs settings, where only around 60% of tasks perform better, leading to degradation in almost half of the tasks (details are shown in Tab. 3 and Appendix). We can reduce the knowledge interference by up to 50%.

This shows that simply scaling the LLM integration does not always improve performance. While adding more models provides a wider range of knowledge, it can also lead to knowledge saturation and inefficiencies. Our Fusion-$\mathcal{X}$ consistently yields better outcomes by effectively selecting and integrating the most relevant models. Therefore, we believe that a selective strategy for LLM integration is crucial. More importantly, designing a better selective strategy can prevent knowledge interference and maximize the overall performance of the fused model.

**Different Model Integration Methods**

We compare Fusion-$\mathcal{X}$ with other integration techniques, including LLM-Blender Jiang et al. (2023), OAssistRM Köpf et al. (2024), UltraRM Cui et al. (2024), and FuseLLM Wan et al. (2024a). Our experiments were conducted on the Big-Bench Hard and MMLU benchmarks.

As shown in Tab. 4, highlight the performance of integrating 4 LLMs with these methods. LLM-Blender (Rank&Fuse) uses a ranker to select the top three results, which are then combined by a fuser, whereas LLM-Blender (Rank) simply selects the top result. Our findings show that our Fusion-$\mathcal{X}$ method consistently outperforms existing works across the evaluated benchmarks. By selecting a robust target model and incorporating specialized models that excel in specific tasks, we achieve superior overall performance. This demonstrates the effectiveness of our targeted fusion approach in creating a more capable and versatile language model.

Table 4: Performance Comparison of Fusion-$\mathcal{X}$ with different integration methods

| Model | BBH | MMLU |
|---|---|---|
| Llama-2-7B | 39.59 | 45.4 |
| LLM-Blender (Rank&Fuse) | 23.58 | 42.1 |
| LLM-Blender (Rank) | 35.66 | 45.3 |
| OAssistRM | 35.21 | 45.7 |
| UltraRM | 37.21 | 46.2 |
| FuseLLM | 40.64 | 46.5 |
| Fusion-$\mathcal{X}$-B | 41.70 | 48.1 |

## 7 CONCLUSION

In this paper, we propose a novel dynamic fusion framework for integrating multiple LLMs. Our adaptive selection network selectively integrates the best-performing source LLMs, overcoming the limitations of existing methods and minimizing knowledge interference. We also introduce a dynamic weighted fusion strategy and a feedback-driven loss function to enhance the fusion process. Our method significantly improves adaptability and performance, offering an efficient solution for LLM integration while maintaining parameter size and computational efficiency. Limitations remain due to the intensive token alignment required prior to training, and future work should explore training on diverse datasets.

## 8 REPRODUCIBILITY STATEMENT

We describe our training settings, dataset used in Sec. 5.1 and Appx. C. We introduce the benchmark used in Appx. D. The details model structure and framework are presented in Sec. 4 and Appx. B.

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
