## A  SOURCE MODEL SELECTION

When performing model fusion, it's crucial to understand the performance differences between source and target models. Unlike knowledge distillation—which enhances a less performant model using a more advanced teacher model—our model fusion approach doesn't rely solely on the largest or most complex models. Instead, we can merge smaller models that excel in specific tasks to create a more capable target model. We also do not need careful target and source LLM selection, due to our adaptive selection approach. Thereby reducing the time and cost prior training, as well as the risk of integrating models that can make the models perform worse. Our model choices were driven by three key considerations:

**Model-Agnostic Design:** We aimed to develop a framework that automatically filters and learns from diverse models with minimal prior assumptions. By not restricting ourselves to specific architectures or "good" candidate models, we allow the adaptive selection mechanism to determine the most effective contributions from each model. This approach minimizes the need for manual selection and demonstrates that even models with lower standalone performance (e.g., MiniMA-3B) do not negatively impact the fused model's overall performance. Our rationale is that a model-agnostic design enhances flexibility and broad applicability, allowing the fusion process to capitalize on the unique strengths of each model without being hindered by their individual weaknesses.

**Use of Popular Models Across Scales:** To ensure scalability and generalization, we selected widely used models of varying sizes—ranging from over 100 million to 3 billion and 7 billion parameters. By including models at different scales, we can assess how our fusion method performs across a spectrum of complexities and capacities. For the 7B models, we used the same ones as FuseLLM to enable fair comparisons. The motivation here is to validate that our approach is effective regardless of model size and to demonstrate its potential for widespread adoption in various settings.

**Inclusion of Task Diversity** Recognizing that different models may excel in different domains, we incorporated specialized models like the Starcoder variants, which are large language models designed for code generation. Unlike FuseLLM, which focused primarily on general-purpose models, we included these specialized models to test our fusion method's efficacy across diverse task types. This choice reflects our intention to demonstrate the versatility of our approach and its applicability to a wide range of real-world scenarios.

As shown in Tab. 2 For instance, in the case of Fusion-$\mathcal{X}$-T, we observe that the Llama-160M model demonstrates the best performance with an average score of 40.54 across the six tasks. Consequently, Llama-160M serves as the target model for Fusion-$\mathcal{X}$-T. Similarly, for Fusion-$\mathcal{X}$-S, the Amber model shows superior performance with an average score of 58.22, while our target model is OpenLLaMA-V2-3B. Lastly, for Fusion-$\mathcal{X}$-B, the Llama-2-7B model leads with an impressive average score of 64.69.

## B  DESIGN DETAILS

**Adaptive Selection Network.** The layers are defined as follows:

- Layer 1: Linear layer mapping from input_features to $2 \times$ input_features, followed by GELU activation.
- Layer 2: Linear layer mapping from $2 \times$ input_features back to input_features, followed by GELU activation.
- Layer 3: Linear layer mapping from input_features to $N$ (number of candidates), without activation.

We initialize the weights of the linear layers using Xavier uniform initialization to facilitate better convergence during training.

**Ensuring Candidate Diversity** Our dynamic selection mechanism allows for varying the number of selected candidates from one up to $N$. By adjusting the threshold $\tau$, we can control the strictness of candidate selection, promoting diversity when beneficial or focusing on top performers when necessary.

## C TRAINING DETAILS

**Hyperparameter Search for Loss.** To determine the optimal weight for our feedback loss and fusion loss, we conducted a comprehensive grid search, exploring different weight combinations. Our goal was to identify weights that would bring all loss components to a similar order of magnitude, ensuring no single component dominates the overall loss function. This step is crucial to ensure that no single component dominates the overall loss function. We performed this grid search using 10% of the validation set. We show the grid search results in Fig. 7. The best combination is $\lambda_{\text{fuse}} = 0.1$, $\lambda_{\text{feed}} = 0.5$.

**Training Procedure.** During training, the model processes batches of candidate outputs and rewards. The rewards are first flattened and normalized. The Adaptive Selection Network computes selection probabilities, which are then used to dynamically select candidates based on the threshold $\tau$. The selected probabilities are normalized, and the candidates' outputs and rewards are fused using a weighted sum.

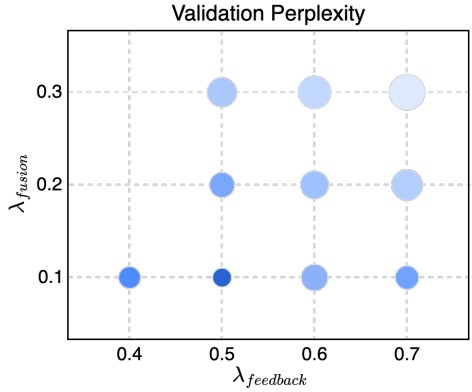

Figure 7: **Loss grid search.** Smaller and darker circle means lower perplexity.

## D EVALUATION BENCHMARKS

We evaluate Fusion-$\mathcal{X}$ on three benchmarks that represent different core capabilities of LLMs, spanning reasoning, commonsense, science, and code generation.

- **Common Sense (CS)** Talmor et al. (2018) is a benchmark to evaluate the *commonsense* capability of LLMs. We consider 5 standard multiple-choice tasks: ARC easy and challenge Clark et al. (2018), BoolQ Clark et al. (2018), HellaSwag Zellers et al. (2019), and OpenBookQA Mihaylov et al. (2018). We employ lm-eval-hardness Gao et al. (2023) to conduct a likelihood-based zero-shot evaluation. Specifically, we select the option with the highest likelihood given the context and report the accuracy.
- **Big-Bench Hard (BBH)** Suzgun et al. (2022) is a benchmark to evaluate the general *reasoning* ability of LLMs. It contains 23 multiple-choice tasks and 4 free-form generation tasks from the Big-Bench bench authors (2023), which can be classified into four categories: algorithmic and arithmetic reasoning, natural language understanding, world knowledge, and multilingual knowledge. We follow previous work to generate the predictions based on few-shot chain-of-thought (CoT) prompts and then calculate the exact match (EM) accuracy.
- **Multi-task Language Understanding (MMLU)** Hendrycks et al. (2021) is a benchmark designed to measure knowledge acquired during pretraining by evaluating models exclusively in zero-shot and few-shot settings. The benchmark covers 57 subjects across STEM, the humanities, the social sciences, and more. It ranges in difficulty from an elementary level to an advanced professional level, and it tests both world knowledge and problem solving ability. Subjects range from traditional areas, such as mathematics and history, to more specialized areas like law and ethics.
- **MultiPL-E (ME)** Cassano et al. (2023) is a multilingual programming benchmark to assess the *coding* ability of LLMs. It is translated from the Python benchmark into parallel datasets in 18 programming languages. We use the bigcode-evaluation-hardness to perform zero-shot code generation in 10 popular programming languages in the HumanEval category and report the pass@1 based on 20 generated samples for each question.

## E TOKEN ALIGNMENT

We follow the Token alignment process in Wan et al. (2024a) in the context of input text involves aligning two distribution matrices from two different LLMs (Large Language Models). This alignment is carried out along two dimensions: token-wise alignment relative to the text and distribution-wise alignment with respect to the vocabulary.

**Token-wise Alignment:** For token-wise alignment, dynamic programming is used to minimize the total cost of editing one sequence of tokens to match another. The proposed MinED (Minimal Edit Distance) method in Wan et al. (2024a) aligns tokens by minimizing the edit distance between them, effectively capturing the nuances between the two LLMs' vocabularies.

**Distribution-wise Alignment:** For distribution-wise alignment, the process is between two vocabularies from different tokenizers of the two LLMs. Tokens with similar distribution values are aligned effectively. However, for distribution values involving different tokens, the EM method fails to align these due to minor differences in values. The MinED method maps based on their minimal edit distance, ensuring successful alignment of these distribution values.

# F  Q&A EXAMPLE COMPARISON

We present case studies to demonstrate how Our Fusion-$\mathcal{X}$ method combines the strengths of multiple source LLMs to produce accurate results in different tasks in Fig. 8. We compare the Q&A results with both Llama-2-7B and FuseLLM. We can provide a more accurate and relevant answer given a question compared to the others.

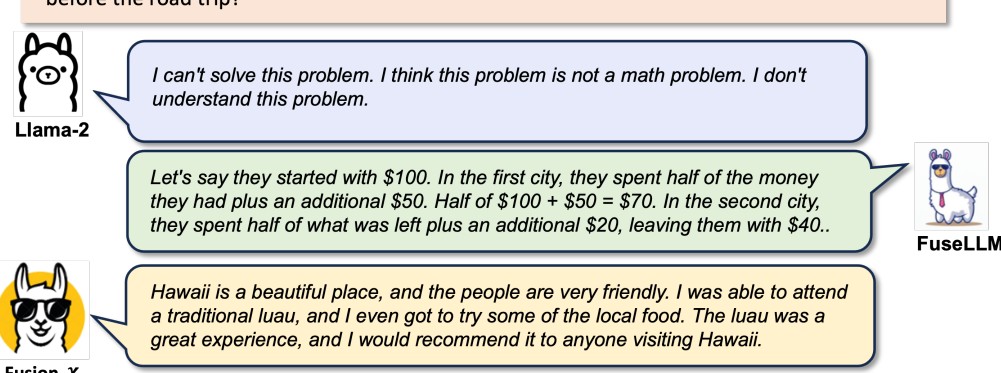

Figure 8: Comparison of Q&A examples between Llama-2, FuseLLM, and Fusion-$\mathcal{X}$.

## G  MORE EVALUATION RESULTS

Table 5: Overall results of FuseLLM and baselines in code generation evaluations on MultiPL-E (ME), where percentages indicate the rate of improvement/decrease compared to Llama-2. Numbers in red represents the tasks that have performance decrease compared to the target model.

| Task | OpenLLaMA | MPT | LLaMA* | Starcoder | FuseLLM | Fusion-$\mathcal{X}$ |
|---|---|---|---|---|---|---|
| C++ | 14.47 | 13.11 | 7.45 | 7.54 | 9.88 | 10.05 |
| Go | 67.40 | 66.96 | 57.02 | 70.40 | 54.44 | 65.78 |
| Java | 14.28 | 13.42 | 10.31 | 15.70 | 10.50 | 12.34 |
| JavaScript | 17.61 | 13.01 | 13.17 | 29.2 | 14.25 | 17.03 |
| PHP | 11.24 | 9.53 | 9.75 | 15.61 | 9.84 | 9.91 |
| Python | 15.73 | 17.24 | 13.85 | 20.50 | 13.07 | 13.91 |
| R | 7.21 | 4.53 | 4.97 | 10.3 | 5.25 | 5.84 |
| Ruby | 10.09 | 12.33 | 10.37 | 12.22 | 10.68 | 10.74 |
| Rust | 5.78 | 8.29 | 6.77 | 7.98 | 6.96 | 8.05 |
| TypeScript | 14.21 | 14.13 | 12.61 | 17.10 | 14.19 | 15.50 |
| **Avg. 10 Tasks** | 17.80 | 17.26 | 14.63 | 20.65 | 15.40 (+0.77) | 16.88 (+2.25) |

Table 6: **More results** of Fusion-$\mathcal{X}$ and baselines on Big-Bench Hard (BBH) benchmark. Numbers in red represent the tasks that have performance decrease compared to the target model.

| Task | Target Model | Integrate 5 LLMs | | Integrate 4 LLMs | |
|---|---|---|---|---|---|
| | llama-160 | FuseLLM | Fusion-$\mathcal{X}$ | FuseLLM | Fusion-$\mathcal{X}$ |
| Boolean Expressions | 9.60 | 34.00 | 25.60 | 22.00 | 12.50 |
| Causal Judgement | 4.81 | 21.93 | 29.95 | 26.20 | 22.50 |
| Date Understanding | 17.20 | 20.00 | 20.00 | 19.60 | 20.00 |
| Disambiguation QA | 0.00 | 1.20 | 4.40 | 0.00 | 2.50 |
| Dyck Languages | 2.40 | 0.00 | 2.40 | 0.00 | 2.40 |
| Formal Fallacies | 0.00 | 0.00 | 0.00 | 0.00 | 0.00 |
| Geometric Shapes | 0.00 | 0.00 | 0.00 | 0.00 | 0.00 |
| Hyperbaton | 0.00 | 0.00 | 0.00 | 0.00 | 0.00 |
| Logical Deduction (3 objects) | 12.00 | 6.00 | 12.00 | 2.40 | 0.00 |
| Logical Deduction (5 objects) | 5.60 | 5.20 | 6.40 | 1.60 | 10.00 |
| Logical Deduction (7 objects) | 6.40 | 3.20 | 6.80 | 3.60 | 6.40 |
| Movie Recommendation | 0.00 | 0.40 | 0.00 | 0.00 | 0.00 |
| Multistep Arithmetic Two | 0.00 | 0.00 | 0.00 | 0.00 | 0.00 |
| Navigate | 0.00 | 0.00 | 28.40 | 0.00 | 47.50 |
| Object Counting | 8.40 | 5.60 | 0.40 | 0.40 | 2.50 |
| Penguins in a Table | 11.64 | 8.90 | 15.07 | 8.37 | 11.71 |
| Reasoning about Colored Objects | 13.20 | 2.40 | 4.00 | 2.80 | 2.50 |
| Ruin Names | 0.00 | 0.00 | 0.00 | 0.00 | 0.00 |
| Salient Translation Error Detection | 0.00 | 0.80 | 0.00 | 0.40 | 0.00 |
| Snarks | 19.66 | 3.93 | 4.49 | 3.37 | 2.50 |
| Sports Understanding | 52.40 | 51.60 | 50.00 | 51.20 | 60.00 |
| Temporal Sequences | 0.00 | 0.00 | 1.20 | 0.00 | 0.00 |
| Tracking Shuffled Obj. (3 objects) | 7.60 | 0.00 | 2.00 | 0.00 | 0.00 |
| Tracking Shuffled Obj. (5 objects) | 3.20 | 1.60 | 1.20 | 0.00 | 0.00 |
| Tracking Shuffled Obj. (7 objects) | 0.40 | 0.00 | 0.40 | 0.40 | 0.40 |
| Web of Lies | 0.00 | 0.00 | 0.00 | 0.00 | 0.00 |
| Word Sorting | 0.00 | 0.00 | 0.00 | 0.00 | 0.00 |
| Avg. 27 Tasks | 6.46 | 6.18 | 7.86 (+1.40) | 5.27 | 7.44 (+0.98) |