# OpenReview forum: "Fusion-$\mathcal{X}$: Advancing LLM Ability with Adaptive Heterogeneous Model Integration"
_ICLR.cc/2025/Conference — Submitted to ICLR 2025_

### Official Review · Reviewer_YWK5 · 2024-10-26

**Soundness:** 3
**Presentation:** 3
**Contribution:** 2
**Rating:** 5
**Confidence:** 5

**Summary:**

This paper proposes a technique to enhance a target LLM by integrating the knowledge of other base LLMs into that model by a fusion process. The scheme they propose essentially performs model fusion by continued pretraining of the target LLM. During this continued pretraining phase, the target LLM is trained using the loss of the standard next-token prediction task and a fusion loss which aims to minimize the cross entropy between the target LLM and some subset of the base LLMs for the same data it is being trained on. The contribution of this paper lies in the selection of the subset of LLMs used where they introduce two components - a learnt selection network to select k LLMS from the pool of base LLMs and a weighted fusion component which produces a weighted combination of the distributions of the k LLMs to be used for the fusion loss. They also introduce dynamic candidate selection which is essentially a threshold on the softmax distribution provided by the selection network to select K most relevant LLMs for that sequence. To prevent
this strategy from collapsing (choosing the same set of candidates everytime), they regularize the outputs such that they are more uniform. The authors present results several ablation studies exploring the design choices behind their proposed scheme, evaluate on several LLMS, and present results on several benchmarks.

**Strengths:**

The main strength of this paper is the extensive evaluation of their technique. The authors evaluate their proposed scheme over several models large models. Given the amount of compute required, their effort is appreciated and I would not ask them to run any further experiments for the sake of size / comprehensiveness. Moreover, the paper is written quite clearly, apart from some notation, and was easy to follow.  The fusion scheme they propose is also interesting though there is limited intuitive explanation of why this proposed scheme should work.

**Weaknesses:**

The main concerns I have this paper are the lack of novelty and motivation.
In terms of novelty, it seems a somewhat simple idea to "learn" the fusion in some way. The terminology used in the paper makes their scheme look a lot more complex than it really is. For instance, Dynamic Candidate Selection is simply thresholded softmax selection using a hardcoded (magic) threshold. The adaptive selection network is a bunch of FFN layers and dynamic weighted fusion is simply selection based on argmax after some softmax selection. The authors also use a loss function to prevent selection collapse and this loss is similar to the loss proposed in [1].

My other main concern is the motivation behind this fusion. If your target LLM is not powerful, it is not going to become as great as the base LLMs. If a base LLM is already pretty good, then it is perhaps better to use it. The authors motivate their work by saying that organizations find it hard to train models given limited data and training resources. But then they do not demonstrate the case where this problem is solved because their results show slightly better performance on tasks which the base models are already pretty good at. For example, Table 3, last line, StarCoder is better on average than all the fused models, so why would we not simply use that model?

I have one more concern about the paper. This is not a weakness but it would be interesting to know the computational overhead introduced by your proposed scheme. If I understand correctly, the input to the adaptive network is a concatenation of probability distributions over model vocabs for a sequence. This is a really huge vector as it is (Seq_Len * Vocab_size) so what exactly are the details for the architecture of the MLPs used?


The notation in the paper is a bit confusing.
Firstly, $i$ is used to index multiple terms, the LLM and the $k$th token being referred to on line 192.
Secondly, the refer to $P_{t}^{\theta}$ as a matrix but denote it as a rows of distributions concatenated together in Line 183.


The citations include author names but not putting brackets around them makes the text look quite confusing.



Typo and Grammar checks
Line 24 - from a flexible number model candidates
Line 82 - , where existing methods > which existing methods


[1] Shazeer, Noam, et al. "Outrageously large neural networks: The sparsely-gated mixture-of-experts layer." arXiv preprint arXiv:1701.06538 (2017).

**Questions:**

Why didn't the authors open source the code? I'd like to be able to reproduce some of their experiments.

How does unifying LLMs solve the data problem? If Org X only has limited data related to their domain, how will using several other LLMs help?

On line 237, it says 'It then computes the logits for each candidate model.' Logits for what? The terminology is used for the logits produced by the LLM (over the vocab) and the for the selection procedure.

How do you deal with LLMs which have different vocabularies?

---

### Official Review · Reviewer_Hf3J · 2024-11-03

**Soundness:** 2
**Presentation:** 2
**Contribution:** 2
**Rating:** 3
**Confidence:** 4

**Summary:**

This paper presents a weighted model merging approach. The weighted model merging approach first trains an MLP-based adaptive selection network to score each candidate model. Then, a threshold $\tau$ is applied to filter out low-score candidate models. The remaining models are merged based on their corresponding scores. These scores are re-normalized after the filtering.

**Strengths:**

1. The score-based weighted merging is a plausible method and likely outperforms uniform weighted merging approaches.
2. The steps of scoring, selecting, and weighting is clear.

**Weaknesses:**

1. The presentation is problematic. Section 4 (methodology) first introduces the adaptive selection network in Section 4.1, but its training pipeline is deferred to Section 4.3. The potential methods explored during the design process in Table 1 are hard to interpret without formally introducing the pipeline. This presentation issue makes the thresholding and weighting operations in Sections 4.2 - 4.3 hard to follow.
2. The design choice of the adaptive selection network is unclear. The weighted model merging problem aims to identify the optimal weight for each candidate model. Here, the input to the adaptive selection network is fixed. The adaptive selection network only takes one input for a given set of candidate models. I'm not sure whether we need to train a network work to make predictions over a single input data point.
3. Identifying the optimal weight is a hyper-parameter optimization problem (HPO). In auto-ML literature, the strong baselines for HPO include Bayesian optimization and bandit. However, these baselines are missing.

**Questions:**

1. Could the authors provide or point me to the results demonstrating that the adaptive selection network can generalize to unseen candidate models? This generalization ability can help distinguish the adaptive selection network-based weighting from standard HPO approaches such as bandit.
2. What is the size of the probabilistic distribution matrices $P$?

---

### Official Review · Reviewer_DQMd · 2024-11-04

**Soundness:** 1
**Presentation:** 1
**Contribution:** 1
**Rating:** 3
**Confidence:** 3

**Summary:**

The paper propose to improve the FuseLLM by introducing an adaptive layer after the fusion of multiple LLMs. Key innovations include an adaptive selection network, which minimizes knowledge interference by choosing the best-performing LLMs, and a dynamic weighted fusion strategy that considers each model’s intrinsic characteristics for more effective knowledge integration. Additionally, a feedback-driven loss function prevents the model from repeatedly selecting the same candidates, promoting flexibility.

**Strengths:**

1. The authors clearly distinguish the new method from the previous in 4 dimensions.
2. Combining multiple models from individual tasks has real-world value, which can easy the pre-training.

**Weaknesses:**

1. The main challenge is not clear and therefore the contribution is vague, i.e., not clear what problem is addressed. The authors discussed many challenges in the 2nd paragraph of the introduction, but none of them have a clear definition, including "simple training", or "interference". What kind of training is simple and why is it important for the fusion method? What is interference? The performance degradation under two different tasks? No citation or definition is ever provided in the introduction. The so-called reduction in interference is also confusing at the end of the introduction. What kind of measure is used? The goal of the work seems to address multiple limitations as shown in Figure 1. But in the contribution, the authors only mention interference and performance improvement by improving interference.
2. The whole method lacks intuition and therefore the paper is hard to follow. What is the intuition behind the proposed method to reduce interference?
3. The proposed method is very incremental, probably because it does not bring in any new insights on the fusion problem itself. The method combines the fusion method, model selection, and feedback-driven loss. Yet, in the ablation study (Sec 6, Table 3), the components are not ablated individually. Thus, it is not even clear how this method work? Which component is the key contributor? Note there is a typo on Sec 6, it should be Table 3 instead of Fig 3.
4. In the ablation study (Table 3), the proposed method only marginally outperforms the FuseLLM. In many tasks, the new method is even worse than the FuseLLM. Without standard deviations, it is hard to justify if the proposed method is better than FuseLLM.

**Questions:**

* What kind of training is simple and why is it important for the fusion method? What is interference?
* What is the intuition behind the proposed method to reduce interference?

---

### Official Review · Reviewer_YWho · 2024-11-04

**Soundness:** 3
**Presentation:** 3
**Contribution:** 2
**Rating:** 5
**Confidence:** 3

**Summary:**

This paper introduces FUSION-X, a dynamic fusion framework for integrating multiple Large Language Models (LLMs) that mitigates knowledge interference. It features an Adaptive Selection Network (ASN) to evaluate and select the best-performing source LLMs based on rewards, allowing for flexible model fusion. A dynamic weighted fusion strategy considers the intrinsic characteristics of candidate models, while a feedback-driven loss function prevents the selector from relying on a fixed set of candidates, enhancing performance. Experimental results show that FUSION-X improves model performance and reduces knowledge interference compared to existing methods.

**Strengths:**

1. FUSION-X achieves an increase in performance across benchmarks by effectively integrating the strengths of various LLMs
2. The framework minimizes the adverse effects of knowledge interference by up to 50%, ensuring that the integrated model's performance remains robust and consistent.
3. FUSION-X's adaptive selection network allows for dynamic model integration, catering to varying tasks and datasets, and maintaining computational efficiency.

**Weaknesses:**

1. The paper highlights the benefits of model fusion but could better specify scenarios where it is essential, such as in domains with limited data or the need for rapid deployment, where integrating specialized small models with large general models could be transformative.
2. Despite the potential of fusing models, the paper does not investigate whether this approach effectively bridges the specific capability gap between these disparate models.
3. The effectiveness of the feedback-driven loss function may be compromised when there is a significant disparity in model capabilities. The paper should address how this method performs under such conditions.
4. Although comparisons are made with LLM-Blender, OAssistRM, UltraRM, and FuseLLM, the paper lacks evaluations against traditional ensemble and weight merging methods, which would provide a more comprehensive assessment of FUSION-X's performance. FUSECHAT was not selected as a baseline for comparison.
5. The discussion on model scaling does not adequately address the integration of newer models like Llama 3.1 8B and Qwen 2.5 into the fusion process.
6. What specific benefits does the dynamic fusion of these smaller models offer over the larger model in terms of performance, efficiency, or adaptability? For example, the advantages of using four 2B models compared to a single 8B model are not clearly articulated.

**Questions:**

See "Weaknesses".

---

### Meta-Review · Area_Chair_fHZK · 2024-12-21

**Metareview:**

This submission introduces a dynamic fusion framework for integrating multiple LLMs that aims to mitigate knowledge interference and improve model performance. The overall rating for this work is negative. Significant concerns were raised about the paper’s lack of novelty and insufficient motivation for the proposed fusion approach. The intuition behind key design choices, such as the ASN and feedback-driven loss, remains underexplored, and several components resemble existing methods with limited innovation. Additionally, the experimental results show only marginal improvements over baselines, raising questions about the practical utility of the framework. The presentation of the methodology could be clearer, and evaluations against traditional ensemble methods or newer baselines were noted as missing.

**Additional Comments On Reviewer Discussion:**

No rebuttal was submitted by the authors

---

### Decision · Program_Chairs · 2025-01-22

Reject